# Implementation of a Multidisciplinary Allied Health Optimisation Clinic for Cancer Patients with Complex Needs

**DOI:** 10.3390/jcm9082431

**Published:** 2020-07-30

**Authors:** Hannah Ray, Anna Beaumont, Jenelle Loeliger, Alicia Martin, Celia Marston, Karla Gough, Shilpa Bordia, Maria Ftanou, Nicole Kiss

**Affiliations:** 1Department of Allied Health, Peter MacCallum Cancer Centre, Parkville, Victoria 3000, Australia; Anna.beaumont@petermac.org (A.B.); Jenelle.Loeliger@petermac.org (J.L.); alicia.martin@petermac.org (A.M.); celia.marston@petermac.org (C.M.); nicole.kiss@deakin.edu.au (N.K.); 2Cancer Experiences Research, Peter MacCallum Cancer Centre, Parkville, Victoria 3000, Australia; karla.gough@petermac.org (K.G.); shilpa.bordia@petermac.org (S.B.); 3Psychosocial Oncology Program, Peter MacCallum Cancer Centre, Parkville, Victoria 3000, Australia; maria.ftanou@petermac.org; 4Institute for Physical Activity and Nutrition, Deakin University, Geelong, Victoria 3220, Australia

**Keywords:** implementation, multidisciplinary, model of care, rehabilitation, complex needs, allied health, cancer patients

## Abstract

This study examined the feasibility of implementing a multidisciplinary allied health model of care (MOC) for cancer patients with complex needs. The MOC in this retrospective study provided up to eight weeks of nutritional counselling, exercise prescription, fatigue management and psychological support. Implementation outcomes (acceptability, adoption, fidelity and appropriateness) were evaluated using nine patient interviews, and operational data and medical records of 185 patients referred between August 2017 and December 2018. Adoption, including intention to try and uptake, were acceptable: 88% of referred patients agreed to screening and 71% of eligible patients agreed to clinic participation. Fidelity was mixed, secondary to inpatient admissions and disease progression interrupting patient participation. Clinician compliance with outcome assessment was variable at program commencement (dietetic, 95%; physiotherapy, 91%; occupational therapy, 33%; quality of life, 23%) and low at program completion (dietetic, 32%; physiotherapy, 13%; occupational therapy, 10%; quality of life, 11%) mainly due to non-attendance. Patient interviews revealed high satisfaction and perceived appropriateness. Adoption of the optimisation clinic was acceptable. Interview responses suggest patients feel the clinic is both acceptable and appropriate. This indicates a multidisciplinary model is an important aspect of comprehensive, timely and effective care. However, fidelity was low, secondary to the complexities of the patient cohort.

## 1. Introduction

Cancer is the leading cause of disease burden in Australia and it is estimated that there will be just under 150,000 new cases of cancer diagnosed in Australia in 2020 [1]. Due to improved detection and treatment, cancer survival has increased, with 68% of Australians expected to survive for at least five years after a cancer diagnosis and in some cancers survival is as high as 90% [2]. It is estimated that in Australia, there are over 1.1 million cancer survivors and this number is expected to increase to 1.9 million by 2040 [2].

A diagnosis of and treatment for certain cancer types comes with a high likelihood of experiencing severe deconditioning, malnutrition, fatigue, distress, loss of function and mental health issues. Cancer-related malnutrition in Australia is reported to occur in 26% to 31% of people with cancer, with particularly high rates of malnutrition among patients with upper gastrointestinal, lung and head and neck cancer [3]. It has been reported that up to 38% of cancer patients experience substantial or severe fatigue, while 33% experience ongoing pain following curative cancer treatment [3,4]. Forty percent of cancer patients experience clinically significant mental health issues, including depression and anxiety [5].

Cancer rehabilitation programs are often designed to deliver the messages of healthy eating, increased physical activity and achieving a healthy weight in line with World Cancer Research Fund recommendations for cancer survivors [6,7]. These programs target patients in the post-treatment phase with a focus on reducing the risk of cancer recurrence, other long-term chronic diseases or further primary cancer diagnosis. Unimodal designs are more commonly used to provide education to cancer survivors, in a group-based setting. Cancer survivors for whom the effects of cancer treatment have resulted in severe physical deconditioning, loss of function, pain, malnutrition, distress and fatigue have needs beyond the scope of a unimodal, group-based rehabilitation program [8,9,10].

Australian and UK survivorship guidelines recommend the use of specialist services for complex (multifactorial) problems arising from cancer treatment [11,12]. These guidelines recognise that this patient group can achieve a substantial benefit from a comprehensive rehabilitation program as an interim step between completion of acute cancer treatment and wellness in survivorship [11,12]. For patients with complex rehabilitation needs, a personalised, coordinated multidisciplinary approach is required to achieve improvements in nutritional status, physical functioning and overall quality of life. Patients should be identified prior to treatment for prehabilitation and streamed directly into rehabilitation during and after treatment. Failure to meet the needs of these patients can have severe consequences to patient outcomes and increase the burden on the health system as demonstrated by multiple national and international evidence-based guidelines [13,14,15,16]. While some studies have examined the impact of a multimodal rehabilitation program provided to patients during or post-treatment, there has been limited research regarding the feasibility of implementing these programs into practice [8,9,10].

A multidisciplinary allied health optimisation clinic was implemented at a tertiary cancer centre in August 2017 for cancer patients with complex needs. It was designed to optimise patients’ condition prior to or during treatment and their readiness for community-based rehabilitation post-treatment. The clinic included a dietitian, psychologist, physiotherapist (PT) and occupational therapist (OT). An individualised structured service was designed to improve physical function, nutritional status, pain and symptom management, fatigue and quality of life. In turn, these improvements would enable patients to resume work and home-based duties, and decrease their burden on carers and the health care system. The clinic design utilised and partnered with existing programs and services providing rehabilitation and survivorship care in the community. This study aimed to assess the feasibility of implementing the optimisation clinic into clinical practice [17].

## 2. Materials and Methods

### 2.1. Design

A mixed-methods implementation study was undertaken at a tertiary cancer centre in Melbourne, Australia, to evaluate the perceived fit and feasibility of the MOC within usual practice [17]. Electronic medical records were retrospectively reviewed and demographic data were collected for participants who accepted appointments to the optimisation clinic between August 2017 and December 2018. Qualitative interview data were gathered from a subsample of participants. The Peter MacCallum Cancer Centre Human Research Ethics Committee approved the study (application no. 18/263R and no. 17/160L), which was performed in accordance with the National Health and Medical Research Council National Statement on Ethical Conduct in Human Research (2007 and updates) and the World Medical Association Declaration of Helsinki of 1975, revised in 2013. A waiver of consent was granted for collection of retrospective data from medical records. All interview participants provided written informed consent before they participated in an interview.

### 2.2. Participants

All patients referred to the optimisation clinic through the period August 2017 to December 2018 were included in this study. All clinicians within the cancer centre could refer a patient to the optimisation clinic prior to, during or post their cancer treatment. Referrals were received by an allied health assistant who subsequently screened patients for eligibility for the optimisation clinic. Patients were considered eligible if they met the criteria for referral for two or more of the allied health disciplines available in the clinic (Figure 1).

At study conceptualisation, only patients with lung or lower gastrointestinal cancers were recruited. These patients were anticipated to have the greatest need. This first phase allowed set up of the clinic to occur on a smaller scale and also allowed us to determine the need of these high-risk tumour streams. Once the clinic processes and capacity had been established, the clinic was then made available to all tumour streams in March 2018, phase two. Patients who were treated in the optimisation clinic were invited by the project manager to participate in an interview after completion of their program.

### 2.3. Model of Care and Implementation Process

The MOC was based upon previous studies reporting on oncology rehabilitation programs, in particular the McGill Cancer Nutrition Rehabilitation (CNR) Program [18]. These studies documented positive outcomes from rehabilitation programs that ranged from eight to twelve weeks in duration, with follow up from clinicians at least once every two weeks. A common aspect of these successful rehabilitation programs was regular multidisciplinary team meetings [8,9,10].

The phase one MOC provided an eight week program, individualised to patient needs, designed to improve physical function, nutritional status, fatigue and quality of life and is described in Appendix A. This MOC operated from August 2017 to March 2018. During this period, the clinicians providing care in the clinic provided feedback that not all patients required the full eight week program, with some ready for discharge from the clinic earlier as patient and clinician goals had been met. In addition, clinicians noted that the screening criteria for referral into the clinic were too restrictive. Therefore, minor changes were made to the MOC to allow flexibility for patients to participate in the program for up to eight weeks based on individual goals or needs, and the screening tool cut off scores required for eligibility to the clinic were adjusted. The phase two MOC operated from April 2018 to December 2018 and is described in Figure 1.

Clinicians involved in the clinic were co-located to facilitate multidisciplinary care and enable participation in a multidisciplinary meeting at the end of each clinic to discuss individual cases. Patients were provided with any combination of individualised nutritional counselling, an exercise program, fatigue management, energy conservation strategies and psychological support, based upon needs identified at the time of screening and during the initial assessment. Additional exercise classes or a home-based exercise program were prescribed by the PT in addition to attendance at the clinic appointments. Patient assessments were completed by clinicians at the commencement and completion of the clinic program in order to guide care in the clinic and facilitate discharge planning upon completion of the program. The assessments used were dependent on the discipline providing care and also guided by the intention of treatment in the clinic, and included: (1) weight and Patient-Generated Subjective Global Assessment (PG-SGA) (dietitian); (2) six-minute walk test, sit-to-stand test, International Physical Activity Questionnaire (IPAQ) and Australian-Modified Karnofsky Performance Scale (AKPS) (PT); (3) Canadian Occupational Performance Measurement (COPM), Brief Fatigue Inventory (BFI), fatigue pictogram and Australian-Modified Karnofsky Performance Scale (AKPS) (OT); (4) Functional Assessment of Cancer Therapy questionnaire (FACT-G) (all patients). Telehealth or phone appointments were offered to patients from regional areas, and to those who were unable to attend appointments in person due to fatigue or a lack of other planned appointments at the cancer centre. At completion of the program, patients were discharged from requiring further allied health management, returned to usual care within the cancer centre or referred to community-based services, depending on their ongoing needs.

### 2.4. Variables and Measures

The success of implementation was evaluated using outcomes derived from the work of Peters et al. [17]. Implementation outcome variables and measures used for evaluation are described in Table 1. Operational data relevant to implementation outcomes were gathered from a clinic specific screening log as well as medical records. Demographic and clinical data were obtained from medical records. Interview data were used to supplement the operational data.

At their final optimisation clinic appointment, all patients who completed the program were invited to participate in semi-structured interviews to determine the acceptability and appropriateness of the MOC. Interviews were audio-recorded and conducted within two weeks of completing the optimisation clinic. Each interview followed a set list of questions and prompts that were designed to understand the perception of the level of support, time commitment, usefulness of the program and ability to make the lifestyle changes recommended (Appendix A).

### 2.5. Statistical Analysis

Descriptive statistics were used to summarise demographic and clinical data. These included counts and percentages for nominal valued variables and medians and interquartile ranges for continuous valued variables. Operational and medical record data relevant to implementation outcomes, including consent to screening and participation in the clinic as well as attendance at scheduled appointments, were summarised using a proportion and 95% confidence intervals; confidence intervals were estimated using the Wilson method [19]. Counts and percentages were used to summarise all other operational and medical record data.

The interviews were audio-recorded and transcribed verbatim. NVivo11 (QSR International, Melbourne, Australia) was utilised for data management and analysis. Interpretive description was used to analyse the data, as this method is responsive to a practice-based discipline like health care, particularly when the results of analysis will be used for the purpose of clinical health services improvement [20]. Analysis was completed by an independent researcher, and double coded by a second independent researcher as a quality check, and any disagreements between the coders were discussed until consensus was reached. Analysis involved reading all the transcripts, generating initial categories then grouping into subthemes of related categories. Subthemes were sorted, synthesised and organised to develop broader themes.

## 3. Results

### 3.1. Study Profile

The demographic and clinical characteristics of the study participants are summarised in Table 2. It is recommended that Table 2 be viewed alongside Figure 2 in order to comprehend the attrition of patients from referral to participation. There were 185 patients referred to the optimisation clinic from August 2017 to December 2018. The median age of patients referred was 64 years, with a slightly higher proportion of females (54%). The dominant tumour streams represented in those referred to the clinic were lung (39%) and colorectal (19%), and the main source of referrals were nurse co-ordinators (38%) and dietitians (23%). These characteristics were consistent across the patients who agreed to screening or were eligible for the clinic. However, on average, patients who agreed to participate in the clinic tended to be younger (median age 60 years) and a higher proportion of females (61%).

### 3.2. Adoption and Fidelity

Adoption, including intention to try and patient uptake, was acceptable, with 88% (162/185, 95%CI: 82%–92%) of referred patients agreeing to screening and 71% (74/104, 95%CI: 62%–79%) of eligible patients agreeing to participate in the clinic. The reasons for declining screening and participation are shown in Figure 2. Thirty-six percent (58/162) of screened patients were deemed ineligible due to requiring input from less than two of the disciplines. Where indicated, patients were referred to the single discipline they required and received usual care within the cancer centre.

Clinic attendance and adherence to the MOC resulted in low fidelity. An individualised program was developed for every patient. However, attendance was poor. Only 41% (30/74, 95%CI: 30%–52%) of patients attended at least 80% of scheduled appointments, with reasons for not attending including, but not limited to, admission to ward, too unwell, fatigue and patients no longer wanting to travel (Figure 2).

Of the 74 patients who agreed to participate, 24% (18/74) failed to commence the clinic program (Figure 2). Of the 56 patients that commenced the clinic program, 57% (32/56, 95%CI: 44%–69%) attended at least 80% of scheduled appointments. Compliance with completion of clinical assessments was variable at program commencement (dietetic, 95%; physiotherapy, 91%; occupational therapy, 33%; quality of life, 23%) and low at program completion (dietetic, 32%; physiotherapy, 13%; occupational therapy, 10%; quality of life, 11%) mainly due to patient non-attendance at clinic appointments.

Clinicians’ adherence to discharge planning procedures at completion of the clinic program was high. Patients requiring ongoing allied health care were referred to community rehabilitation (*n* = 16) or returned to usual allied health care within the cancer centre (*n* = 10).

### 3.3. Acceptability and Appropriateness

Nine patients were interviewed on completion of treatment in the optimisation clinic. Interviews ceased when data saturation was achieved, as identified by the interviewer that no new information was emerging. Interview participants were younger (median age 53 years), with a higher proportion of females (67%) compared to all patients who participated in the clinic. The dominant tumour streams of those interviewed is similar to all clinic participants with 32% lung cancer patients and 19% colorectal cancer patients.

Analysis of patient interviews identified six inter-related themes: integration, individualised care, quality of care, convenience, multidisciplinary care, and model in evolution.

#### 3.3.1. Integration

Patients felt that the clinic facilitated integration through partnership with community-based health programs, access to a multidisciplinary team of allied health clinicians who provided them with prompt and individualised care, and incorporation of concurrent appointments, saving time and providing them with a broader perspective. Patient ID07 (F, 36 yo) said “It seems to be quite a seamless experience, you know, it’s no waiting around, it’s actually been a really great thing”. Whilst patients appreciated that integrating multiple appointments in one day was good in theory, they realised that this sometimes resulted in missed appointments due to the previous appointment running overtime, or ineffective appointment flow in the clinics.

#### 3.3.2. Quality of Care

Quality of care was an important theme across all patient interviews. Most patients spoke highly about the care received. Patient ID03 (M, 62 yo) said “everybody seemed very concerned and made sure I was on the right path” and patient ID05 (F, 46 yo) said “Yeah I, um, yeah I couldn’t speak more highly of them and I feel very fortunate that I had my treatment there and that I engaged in it for sure”. The quality of care was evident both in their consultations and treatment plans. Patient interviews (*n* = 7) highlighted that the clinicians were perceived as supportive and involved. Overall, patients found the team professional and supportive, and the program sustainable. Patient ID04 (M, 74 yo) said “I felt as though I was somebody special” with a further three patients contributing similar quotes to support this theme.

Patients (*n* = 7) reported that the clinic facilitated changes in their lifestyle which improved their physical and functional wellbeing. Patients felt that their exercise and activity levels improved, changes to diet had a positive impact on their health, and new hobbies helped with focus and engagement. Some patients (*n* = 6) highlighted that they were able to continue with the lifestyle changes that resulted from participating in the clinic. On the other hand, some could not maintain the changes due to barriers like fatigue (*n* = 1), lack of motivation (*n* = 1), inability to recall advice (*n* = 1) and emotional breakdown (*n* = 1).

#### 3.3.3. Convenience

Patients’ interview responses suggested that they found the clinic model convenient and accessible (*n* = 4). Patient ID05 (F, 46 yo) said “I believe it was as easy as it possibly could be”. With access to a multidisciplinary team of allied health clinicians, patients were able to streamline appointments and minimise time spent at the hospital. Where appropriate, patients could discuss their care needs with multiple clinicians simultaneously, reducing the need to constantly repeat themselves and resulting in an effective and productive consultation (*n* = 3). Patient ID09 (F, 33 yo) said “Some days when I was really tired, the physiotherapist and OT saw me at the same time. They were all asking similar questions so it saved me having to repeat myself to them separately. The clinic worked really well in that way”. Most patients (*n* = 7) felt that the appointment time was sufficient and appointment flow in clinic was reasonable, with no excessive waiting times. One patient stated “It made it a lot easier, knowing all the appointments were together. I didn’t have to come into the hospital numerous times, you’re already coming into the hospital for other appointments, so it made it much easier knowing you were coming in for all your appointments” ID09 (F, 33 yo). A further three patients made similar quotes supporting this theme.

Some patients acknowledged and appreciated the fact that the clinic incorporated telehealth services for patient convenience, preventing patients from travelling further or waiting longer to access care. Although one patient felt that clinicians also need to remember that face-to-face consultation cannot be replaced by technology, especially for appointments that require a more hands on approach.

#### 3.3.4. Multidisciplinary Care

The multidisciplinary structure of this clinic was its greatest difference in comparison to usual care and patients emphasised the benefits of this feature (*n* = 6). Patient ID01 (F, 54 yo) said “being that everyone was together, that made it much easier to get the help that you needed” and patient ID05 (F, 46 yo) said “Look I would definitely recommend it wholeheartedly to everyone that’s having treatment because they really did seem to work as a team”. Patients felt that due to the complexities arising from their diagnosis and treatment, they often end up seeing a range of specialists to address their treatment and care needs. One patient ID05 (F, 46 yo) commented that they felt fortunate being engaged with the clinic and called it a “one stop shop.” Patients (*n* = 2) also identified that the clinic facilitated clinician communication which not only led to prompt action but also allowed patients to receive vital information from multiple perspectives at the same time.

Whilst most patients’ responses suggested that the team-based care approach was clearly evident, one patient who attended the clinic at its commencement felt differently, stating “Yeah I didn’t catch the feeling of the team” ID04 (M, 74 yo). This patient’s feedback is discussed further in the ‘model in evolution’ theme below.

#### 3.3.5. Individualised Care

Patients felt that every patient’s needs and care requirements are unique and so should be considered, adapted and catered for throughout their care. Patients acknowledged that the clinicians in the optimisation clinic assessed and addressed their needs as an individual and tailored the program to them. Patient interviews reflected that upon commencement of the program they might have opted to visit certain clinicians, but during their involvement in the program, their needs were assessed on an ongoing basis and they were referred to clinicians accordingly. Patient ID01 (F, 54 yo) said “Like one day there I think I’d lost a bit of weight and they say you should see the dietitian and then they set it up and I was able to go directly across to see”.

#### 3.3.6. Model in Evolution

Patients felt that the model of the clinic was a great concept but has room for improvement. One patient ID04 (M, 74 yo) commented on their experience in being involved in the clinic upon its commencement as “And, yeah, coming to it was a bit like somebody building a house, I think well the frame’s up but there’s no roof on. And the next time said well the frame’s up and the roof’s on but there’s no walls. And so a bit, um, yeah not, you know, as though it wasn’t quite ready or something, yeah, but that’s it.”

## 4. Discussion

Comprehensive rehabilitation following a diagnosis of cancer requires a personalised, coordinated multidisciplinary approach. This study assessed the feasibility of such an approach using a structured MOC developed by the multidisciplinary team. Overall, the MOC was shown to be a feasible means of delivering cancer rehabilitation with high adoption, acceptability and appropriateness, albeit with challenges related to fidelity and adherence.

The optimisation clinic was designed for patients with complex needs, defined as those requiring the services of two or more allied health disciplines who were identified through a screening process prior to acceptance into the clinic. Patient adoption of the clinic, with regards to willingness to participate in the screening process and ultimately the clinic program, was good (88% and 71%, respectively) and higher than participation rates reported in studies of similar rehabilitation programs [9,10]. However, following agreement to participate, almost a quarter of patients failed to commence the program and only just over half completed the program, demonstrating relatively low fidelity. A high drop-out rate is not unexpected when targeting patients with complex needs, particularly when considering that patients meeting this criteria were deconditioned and likely to be those with more advanced disease. These figures are consistent with similar cancer rehabilitation clinics designed for patients with advanced cancer, where 20% of patients fail to commence the program and drop-out rates ranging from 30% to 42% have been reported [9,10]. Furthermore, these studies report similar reasons for patients failing to complete the rehabilitation program including disease progression, death, hospitalisation, loss of interest, patients too busy or feeling too unwell [9,10,21]. The complexity of these patients and the drop-out rate is a factor that must be anticipated, requiring flexibility in the MOC such as the use of telehealth where feasible. As previously stated, the outcome variables for this study were derived from Peters et al., who define fidelity as ‘the degree to which an intervention was implemented as it was designed in an original protocol, plan, or policy’ [17]. In line with our aim, this study assessed fidelity of the MOC rather than the individual clinicians’ intervention (e.g., adherence to exercise prescription). We acknowledge that expanding the criteria for assessing fidelity would have added value to this study, and identify this as a limitation. While the focus of this study was on feasibility, studies of similar programs have demonstrated improvements in nutritional status, physical function and fatigue [9,10], indicating the importance of providing these models of care despite some of the challenges. These health benefits are reflected in patient views expressed during the qualitative interviews.

Further contributing to the challenges with fidelity was low clinician adherence to aspects of the MOC, particularly completion of clinical assessments. The clinical assessments were included in the MOC in order to individualise and guide care both during and upon discharge from the clinic program. A number of factors contributed to difficulties in completing these assessments, including patient drop-outs and patients not attending in person for the final appointment. Telehealth appointments were used for patients unable to attend in person where feasible. However, many of the clinical assessments are designed to be completed face to face and were unable to be completed through telehealth. Telehealth and other technology platforms are a growing field in order to increase capacity to deliver broad reaching interventions, particular to those in regional areas. Systematic reviews in people with cancer demonstrate technology-supported interventions have beneficial effects on health behaviours and outcomes [22,23]. Future iterations of the optimisation clinic may need to adopt increased use of technology, and subsequently consider suitable clinical assessments to be used via these platforms. Understanding and evaluating clinician behaviour change could have also improved clinician adherence to the MOC and is considered an important way of measuring success of a complex intervention model. Application of the Theoretical Domains Framework by Michie et al. to understand behavioural factors that served as enablers or barriers to implementation factors would be a valuable strategy to use in future studies [24].

Clinician engagement is a key factor for successful implementation of changes in health service delivery and development of new health care models [25,26]. While a formal co-design methodology was not followed, engagement of key stakeholders including oncologists, allied health clinicians and nurses began at the outset of developing the MOC to ensure representation of key users and referrers to the clinic. This approach has been successfully used in the development of other models of care in the same cancer centre [27]. However, as early as the planning phase, there was a lack of oncologist involvement in steering committee meetings and in provision of feedback during the development of the MOC. A number of strategies were employed to improve engagement, including direct contact with committee members, scheduling one-on-one meetings and providing opportunities to provide input via email. Despite this, lack of engagement was a challenge encountered throughout the development and implementation of the optimisation clinic and was evident in the lack of referrals received from oncologists. A key difference between the optimisation clinic and the McGill cancer nutrition rehabilitation program upon which the clinic was modelled, was the inclusion of a physician, nurse and clinic coordinator in the clinic team [9,10]. These key team members are suggested to assist in building trust, facilitating communication, supporting patient navigation of hospital systems as well as managing the clerical and administrative work required to operate the clinic [28]. Other than a project officer to support the development and implementation of the clinic, the clinic was required to operate through reallocation of existing allied health resources as no additional medical and nursing resources were available.

Patient interviews indicated that the acceptability and appropriateness of the clinic were high. Patients acknowledged the range of allied health specialists required to address their complex needs, and interview responses suggested the multidisciplinary clinic model was vital in providing comprehensive, timely and individualised care. This is consistent with previous research that has demonstrated improved timeliness of care from multidisciplinary clinics in cancer care. Existing literature, however, largely relates to multidisciplinary clinics of physicians, and timeliness of care refers to decreasing the interval from diagnosis or first consultation, to commencement of treatment [29,30,31]. In our clinic, patients perceived the care received as high quality and able to facilitate positive lifestyle changes. This is similar to qualitative studies of multidisciplinary physician clinics where increased patient satisfaction, increased collaboration and appreciation of patient-centred care have been reported [32,33]. Overall, patients appreciated the integrated team-based approach to their care. However, from a patient perspective, a number of opportunities for improvement were identified, most of which were consistent areas for improvement identified by the project team. In particular, these included addressing barriers to patient involvement in the clinic program such as fatigue, low motivation and information overload. The greater utilisation of telehealth and additional technologies such as web-based or text message support may require consideration to overcome these barriers. In addition, some patients attending the clinic early after its implementation recognised the clinic was new and an evolving model.

It is acknowledged that there are some limitations of the study. Patient interviews represent the views of a small number of patients and we were unable to capture the views of those who did not commence or complete the program. While it was not the focus of this study, the impact of the MOC on individual outcomes was not assessed and therefore we are unable to determine the effectiveness of the care provided on patient outcomes. A strength of this study is the comprehensive assessment of the feasibility of implementing this type of model into usual care.

## 5. Conclusions

This study demonstrates the feasibility of implementing a multidisciplinary allied health optimisation clinic designed to improve fatigue, nutritional and functional status. The optimisation clinic facilitated the coordinated and team-based care of people with cancer with complex needs. However, a number of opportunities for improvement were identified, including further consideration of flexible, potentially technology-supported approaches to care delivery. While patient outcomes were not assessed, improvement in health outcomes were perceived by patients.

## Figures and Tables

**Figure 1 jcm-09-02431-f001:**
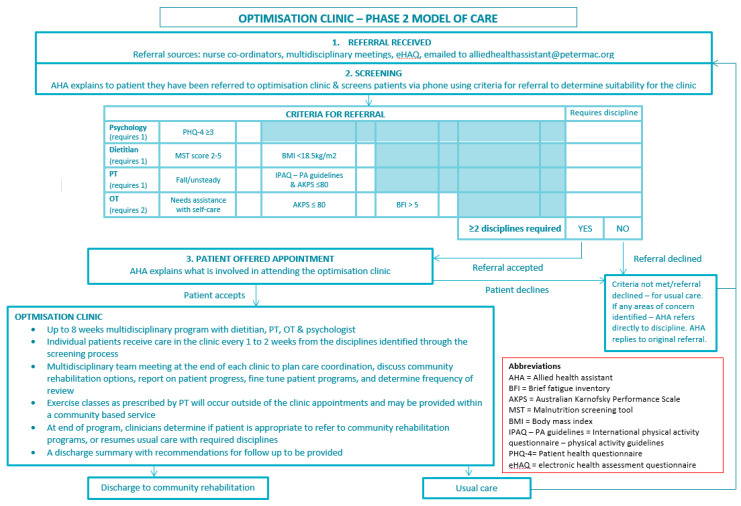
Phase Two Optimisation Clinic Model of Care *. * Refer to Appendix A to identify the modifications made to the model of care (MOC) between phase one and two; changes included program length and criteria for referral.

**Figure 2 jcm-09-02431-f002:**
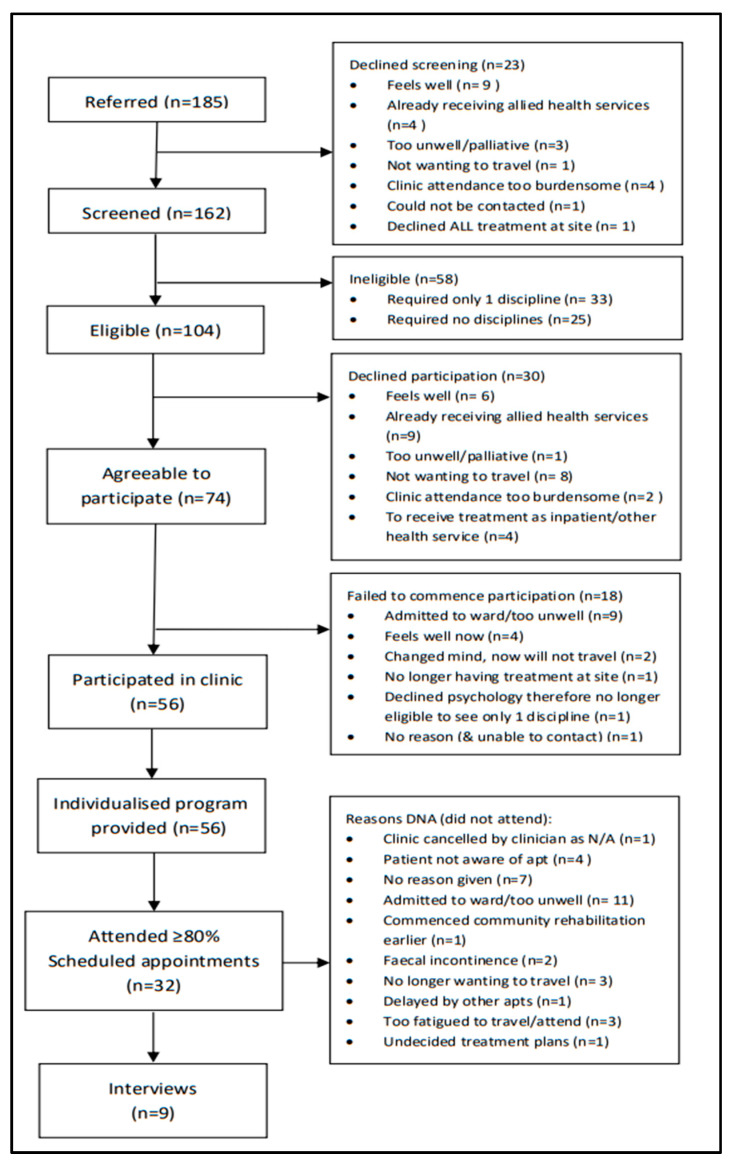
Participation Flow Diagram.

**Table 1 jcm-09-02431-t001:** Implementation outcome variables.

Outcome	Aspect	Measure
Acceptability	Satisfaction	Interview data
Adoption	Intention to try	Operational data: consent rate for screening, reasons for declining screening
	Patient uptake	Operational data: consent rate to participate in clinic, reasons for declining clinic participation, reasons for ineligibility
Fidelity	Attendance	Medical record data: attendance at scheduled appointments, reasons for non-attendance
	Adherence	Medical record data: completion of assessments by clinician, rate of community referral at program completion, delivery of individualised program
Appropriateness	Perceived fit	Interview data

**Table 2 jcm-09-02431-t002:** Characteristics of patients referred to the optimisation clinic, those who agreed to screening, those who were eligible for the clinic and those who agreed to participate in the clinic.

Characteristics	All Referred (*n* = 185)		Agreed to Screening (*n* = 162)		Eligible (*n* = 104)		Agreed to Participate (*n* = 74)	
	*n*	%	*n*	%	*n*	%	*n*	%
Age (in years)								
Median	64		64		64		60	
Interquartile range	55 to 71		53 to 70		51 to 70		49 to 69	
Range	19 to 93		19 to 93		21 to 92		21 to 86	
Sex								
Male	85	46	72	44	44	42	29	39
Female	100	54	90	56	60	58	45	61
Tumour stream								
Breast	11	6	11	7	11	11	7	9
Colorectal	36	19	30	19	19	18	14	19
Gynaecology	19	10	17	10	9	9	7	9
Haematology	16	9	15	9	9	9	9	12
Head and neck	3	2	3	2	3	3	3	4
Lung	73	39	61	38	37	36	24	32
Sarcoma	12	6	11	7	6	6	5	7
Skin/melanoma	7	4	7	4	5	5	2	3
Upper gastrointestinal	2	1	2	1	2	2	2	3
Urology	6	3	5	3	3	3	1	1
Treatment								
Chemotherapy	100	54	92	57	58	56	42	57
Chemotherapy and radiotherapy	24	13	19	12	13	13	11	15
Declined treatment	1	1	1	1	1	1	0	0
Nil (surveillance only)	11	6	10	6	8	8	8	11
Nil (too palliative)	2	1	2	1	2	2	0	0
Radiotherapy	28	15	22	14	11	11	6	8
Surgery	19	10	16	10	11	11	7	9
Distance from hospital								
Median	24		25		25		24	
Interquartile range	12 to 49		13 to 49		12 to 55		10 to 55	
Range	1 to 862		1 to 862		1 to 862		1 to 862	
Source of referral								
Dietitian	43	23	35	22	19	18	15	20
Doctor	20	11	19	12	13	13	11	15
e-HAQ(electronic health assessment questionnaire)	21	11	19	12	14	13	5	7
Nurse	8	4	8	5	5	5	3	4
Nurse coordinator	71	38	60	37	36	35	27	36
Occupational therapist	5	3	5	3	3	3	3	4
Physiotherapist	13	7	12	7	10	10	8	11
Psychologist	2	1	2	1	2	2	1	1
Social worker	2	1	2	1	2	2	1	1

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
