# Peer review of "Implementation of a Multidisciplinary Allied Health Optimisation Clinic for Cancer Patients with Complex Needs"

_jcm, 2020, doi:10.3390/jcm9082431_

Round 1
Reviewer 1 Report
The authors implemented and evaluated a multidisciplinary allied health optimization clinic for cancer survivors. Overall, this is a well-written article that addresses an important area of cancer care that is often neglected.
The authors use both the acronyms MDT and MOC, though aren't consistent with their use of MOC as demonstrated in first paragraph of Discussion where they describe a "structured model of care". I suggest using one acronym to minimize confusion.
There would be value in adding more discussion, either in Introduction or Discussion, around the existing literature related to multidisciplinary clinics in cancer care. Multidisciplinary clinics have been shown to improve timeliness of care, and qualitative studies have also been performed to assess the patient and caregiver experience. How do the results of this study compare with existing literature? What opportunities do they present for how this MDT MOC could be improved?
Is an author missing from the listed authors? The author list does not include a name after "and 4".
Table 2 is somewhat hard to read with several columns of data and lack of clear deliniation between sections. I suggest converting some of this data to Figure format to pictorally demonstrate the attrition of patients from referral to participation.
It would be ideal to include more quotations of patients from the interviews, and comment on how many patients contributed quotes that supported each theme. This would strengthen the qualitiative component of the study.
Author Response
Thank you for your time and expertise. The feedback you provided is very encouraging and we appreciate your responses to assist us to improve our manuscript. We have acknowledged your responses and amended our work accordingly.
Please see the attached document for details of each modification made.

Reviewer 2 Report
Review of Manuscript# jcm-862115 (Journal of Clinical Medicine) – July 13, 2020
Title: Implementation of a multidisciplinary optimisation clinic for cancer patients with complex needs
This manuscript describes the feasibility of implementing a multidisciplinary team (MDT) model to care for cancer patients with complex needs. The patients in this study were provided up to eight weeks of nutritional counseling, exercise prescription, fatigue management, and psychological support. Using a mixed-methods approach, the authors analyzed implementation outcomes, namely, acceptability, adoption, fidelity, and appropriateness, based on data collected from 185 patients referred between July 2017 and December 2018. The authors reported that while acceptability, adoption, and appropriateness were robust, the fidelity outcome was mixed and less than desirable.
Overall, the manuscript is very well written and has the potential to make an important contribution to the debatable literature on multidisciplinary care models for cancer patients. However, the manuscript has a few conceptual and organizational issues that need to be considered and addressed. The main conceptual issue is the phrase used to describe the multi-D clinic in the study. Usually the notion of multidisciplinary clinic involves various physician specialists. For example, a lung cancer multidisciplinary clinic comprises: primary care physicians, hospitalists, emergency room physicians, pulmonologists, medical oncologists, radiation oncologists, thoracic surgeons, palliative care physicians, and a nurse navigator. Since none of these specialities were involved in the multi-D implementation in this study, it is more suitable to call it an “Allied Health Multidiscipinary Optimization Clinic” or “Allied Health Optimization Clinic.” The authors should consider using one of these two phrases or another similar phrase throughout the mansucript to aptly describe the nature of the multi-D clinic described in this study.
Please see below my detailed comments:
Line 3: This multi-D clinic is primariy focused on Allied health. Therefore, the title should reflect it.
Line 6: There is a misplaced “and” at the end of the line.
Line 12: The comma at the end of line after “study” should be deleted.
Line 23: Given the limited success with adoption, the authors may want to soften the sentence regarding the success of the adoption of the optimisation clinic.
Line 26: There should be a comma after “fidelity was low”.
Line 28: Consider adding “allied health” and “cancer patients” in the keywords.
Line 82: How were the patients selected for participation in the qualitative interviews? The sample size seems small, even though the author indicated that saturation was reached with only 9 respondents.
Line 90: Make study period consistent throughout the mansucript. In the abstract it says, August 2017 to December 2018 but on line 90 it says, July 2017 to December 2018.
Line 96: Provide a brief explanation on why only patients with lung or lower gastrointestinal cancers were recruited in the first phase of the study. Also, the authors may want to be more explicit about having two phases in the study.
Page 4 and page 12: There are two iterations of the same figure S1. They appears to be duplicate. The second one on page 12 seems more refined. Please look into it. Also, explanations for some of the abbreviations, such as PMCC, MDMS, etc. are missing.
Line 113-116: What changes were made to relax the criteria for eligibility of patients during the second phase of the study?
Line 140: There should be a period for et al.
Line 147: Please check the time period (from December 2017 to May 2018). It does not appear to be correct.
Line 150: Address the duplication of Figure S2 (one on page 12 and the other on page 13). Also mentioned earlier in my comments.
Line 151: There are some spacing issues in Table 1.
Line 179: Table 2 has some formatting issues. See column 4 (Eligible).
Line 180: The authors discussed Adoption and Fidelity in this section. However, the criteria for assessing the fidelity is minimal and should be expanded further.
Line 200: Please rephrase this sentence. It does not read well.
Line 202-203: Again, it says Figure 2. Please unduplicate and correct the figure numbers both in text and in the illustrations throughout the manuscript. Also, the Participation Flow Diagragram is not legible. Please present a legible diagram.
Line 213: The six inter-related themes (integration, individualised care, quality of care, convenience, multidisciplinary care, and model in evolution) should be placed at the second level of headings and not at the same level as “Acceptability and Appropriateness.”
Line 248: It should read “Multidisciplinary care”
Discussion: The authors should be cautious in presenting the level of success at various stages of implementation because only about 31% eligible patients ended up participating and attended 80% of the scheduled appointments. Similarly, there was a very high rate of non-completions of assessments. The authors may also want to discuss the strategies that could enhance success of various implementation outcomes in the future.
Author Response
Thank you for your time and expertise. We are grateful for the feedback you provided and we appreciate your responses to assist us to improve our manuscript. We have acknowledged your responses, including your concern above, regarding the phase used to describe the multidisciplinary, and have amended our work accordingly.
Please see the attached document for details of each modification made.
